# Improved Methodology of Cross-Sectional SEM Analysis of Thin-Film Multilayers Prepared by Magnetron Sputtering

Malwina Sikora [1,2], Damian Wojcieszak [1,*], Aleksandra Chudzyńska [3,4] and Aneta Zięba [1,2]

1 Faculty of Electronics, Photonics and Microsystems, Wrocław University of Science and Technology, Janiszewskiego 11/17, 50-372 Wrocław, Poland
2 Nanores Company, Bierutowska 57-59, 51-317 Wroclaw, Poland
3 Compact X Company, Bierutowska 57-59, 51-317 Wroclaw, Poland
4 Institute of Low Temperature and Structure Research, Polish Academy of Sciences, Okólna 2, 50-422 Wroclaw, Poland
* Correspondence: damian.wojcieszak@pwr.edu.pl

**Abstract:** In this work, an improved methodology of cross-sectional scanning electron microscopy (SEM) analysis of thin-film Ti/V/Ti multilayers was described. Multilayers with various thicknesses of the vanadium middle layer were prepared by magnetron sputtering. The differences in cross sections made by standard fracture, focused ion beam (FIB)/Ga, and plasma focused ion beam (PFIB)/Xe have been compared. For microscopic characterization, the Helios NanoLab 600i microscope and the Helios G4 CXe with the Quanta XFlash 630 energy dispersive spectroscopy detector from Bruker were used. The innovative multi-threaded approach to the SEM preparation itself, which allows us to retain information about the actual microstructure and ensure high material contrast even for elements with similar atomic numbers was proposed. The fracture technique was the most noninvasive for microstructure, whereas FIB/PFIB results in better material contrast (even than EDS). There were only subtle differences in cross sections made by FIB-Ga and PFIB-Xe, but the decrease in local amorphization or slightly better contrast was in favor of Xe plasma. It was found that reliable information about the properties of modern nanomaterials, especially multilayers, can be obtained by analyzing a two-part SEM image, where the first one is a fracture, while the second is a PFIB cross section.

**Keywords:** cross section; preparation techniques; SEM; FIB/Ga; PFIB/Xe; thin-film materials; multilayer; magnetron sputtering

## 1. Introduction

Scanning electron microscopy (SEM) allows for the study of a broad range of specimens for scientific and industrial purposes. In addition to metal alloys [1], biological and geological samples [2–8] and polymers [9–11] can also be investigated, as well as electronic devices [12,13] and mechanical components incorporated into larger structures [14]. Advanced imaging techniques require the use of an appropriate method for sample preparation. Nevertheless, preparation for SEM is still not a trivial task, especially when the characteristic dimensions of the objects are at the nanometric level. The choice of the appropriate technique depends on the nature of the sample, including its conductivity, state of aggregation (liquid, solid), and form (powder, bulk material, thin film, etc.). The preparation is closely related to the type of information that should be obtained from the sample with the aid of SEM. The smaller the area of interest, the more demanding sample preparation is, not to mention the need to reduce artifacts and unwanted modification of the sample itself. Selection of the appropriate preparation technique, even from among the already recognized methods, requires knowledge of how each of them changes the actual properties of the sample. Making such a seemingly simple choice is not easy due to the very small number of studies that compare the results of different methods of preparation of the

same samples [15,16]. In the case of modern nanomaterials, this is also difficult because the parameters that describe them are often statistical in nature, such as the average size of the crystallites [17]. For this reason, a good research material for a reliable comparison of several methods is multilayer coating, where a well-defined single layer of the desired thickness can be buried at a specific and known depth [18–23]. It should be noted that a series of multilayers with a gradually modified thickness of selected layers allows for better visibility of quite subtle differences and artifacts that could have arisen, especially in the interface area [24–26]. Another issue that makes it difficult to choose the optimal preparation is related to the fact that most publications refer to the advantages and disadvantages of one single method, and unfortunately often without any explanation of the reasons for its choice [27–33]. The third issue to be mentioned is the use of quite contrasting materials (for SEM), which will have a rather poor application for materials consisting of elements with a similar atomic number, such as the titanium and vanadium selected for our research [34,35]. Their use results in a very low electron density contrast at the interface and is an additional difficulty in SEM imaging, but will allow for the elimination of much more subtle artifacts.

An important issue of SEM preparation, especially in the context of imaging advanced nanostructures, is the manufacturing of cross sections. The selection of an appropriate technique, especially in the case of multilayer coatings, is key to the correct visualization of their properties and to limiting the possibility of introducing artifacts. Imaging of advanced electronic components, where characteristic dimensions are at the nanometer level, requires working with cross sections devoid of as many artifacts as possible. Modern electronic or optoelectronic systems usually have a multilayer structure, which results in the need to characterize them at the level of single, nanometric films. For this reason, thin-film materials (in the form of single films or multilayers) are an important group where advanced preparation techniques for SEM analysis are needed. They should include not only unchanged microstructure, surface roughness, or thickness, but should also give true information about the material composition. Nowadays, it is possible to characterize a single film included in the multilayer structure with thicknesses of at least tens of nanometers [36]. Multilayer coatings (used as optical filters), which are a stack of high and low refractive index layers, have thicknesses of individual layers often <10 nm. Similarly, the characteristic dimensions have transistors, which are now produced with the so-called 7 nm technology. Therefore, these factors have led to the development of electron microscopy. Currently, the newest apparatus provides a resolution of ca. 1.4 nm. Therefore, it seems sufficient to visualize various types of advanced electronic, photonic, or optical systems based on nanostructures. However, there are still many problems with the proper preparation of the samples for SEM. It should be noted that while there is no ideal method that would be suitable for every sample, an improved methodology based on the hybrid preparation technique as a multistep approach can be applied as a modern solution.

Improvement of the SEM results requires the use of such cross section preparation techniques that will maintain the real properties of individual layers and interfaces [16,37]. There are a number of methods for manufacturing cross sections of thin films. Among these are the break method [37–39], the pre-cut technique [40], ultramicrotomy [37,41,42], grinding and polishing preceded by resin encapsulation [37], as well as ionic techniques: ion polishing [43–45] and a focused ion beam [46,47]. In each method, there are artifacts that affect the visualization of the properties of the samples, which one needs to be able to dissect. In the literature on thin films, the field of preparation is usually ignored; only SEM images of cross sections are presented. Many works, e.g., [48,49] show results obtained by only one technique, which usually is a standard fracture. There is a lack of summaries comparing different preparation methods with each other, and this paper is an attempt to address this niche. In our opinion, the improved methodology for the manufacturing of thin-film preparations for the purposes of SEM research can be successfully implemented using three methods, which are fracture and focused ion beams with gallium ion source and xenon plasma.

Standard fracture is a very common method. It does not affect the microstructure of the samples, enhances their morphology, and is cheap and fast. However, its main disadvantage is low repetition and high susceptibility to accidental damage. Delicate samples, e.g., coatings deposited on polymer substrates, may require preceding fracture by freezing a sample in liquid nitrogen in order to obtain brittle fracture of the polymer [47].

An ion technique used for the cross section of coatings is the focused ion beam (FIB) [46,47]. The structure and chemical composition of multilayer structures can be best visualized when imaging occurs perpendicularly across the interface [16], hence the ability to perform cross sections without artifacts becomes crucial. FIB enables microstructural characterization of coatings by cross-sectioning and preparation of specimens for scanning purposes, as well as for transmission electron microscopy [48–50]. The most important FIB advantage is the fabrication of the specimen in a selected area of interest. Compared to mechanical polishing, it avoids deformation, streaking of polished layers, and filling of existing cracks [48]. The duration of such a FIB preparation is usually not longer than 15 min. The cross section is formed by multiple passes of a high-energy ion beam (with a current value of ca. 2.7 nA). The penetration depth gradually increases, and at the deepest point of the section, its surface is perpendicular to the surface of the sample. The coarse-picking stage is followed by polishing the cross section with the ion beam obtained at a lower current (about 0.15 nA), after which the sample is ready for imaging. Iterative polishing and imaging of the cross-sectional surface enable a three-dimensional reconstruction of the microstructure [48].

It should be noted that in many works (e.g., [36]) dedicated to SEM studies of thin film materials, especially those prepared by PVD or CVD methods, the methodology for preparation techniques for accurate microscopic visualization is not given or has a residual description. This causes difficulties in the proper interpretation and characterization. Hence, there arises the need to develop a complex methodology. The aim of our work was to develop an improved methodology for the study of thin-film coatings on the basis of known and existing methods, including the aforementioned FIB. This innovative approach does not concern the improvement of standard methods, but it is devoted to a multithreaded approach to the SEM preparation itself, which will allow us to retain information about the actual microstructure and ensure the best possible material contrast (even for elements with similar atomic numbers). The research performed on the example of Ti/V/Ti multilayers, in order to present the sense of using such an improved methodology, showed that the true information about the tested sample can be obtained by assessing its microstructure based on SEM images made using the fracture technique, while the material composition can be well visualized using FIB methods (better than by EDS). The literature review also indicates the lack of data that describe the application of such a methodology to the analysis of multilayer coatings and, in particular, its effect on their structure. Comprehensive studies comparing the use of different techniques for cross section preparation, especially using two different sources of focused ion beam (including focused xenon plasma) are also omitted. Therefore, this work fills that niche.

## 2. Materials and Methods

### 2.1. Multilayer Project

Ti/V/Ti multilayer structures were designed for the purpose of the present study. Their construction and elemental composition were chosen for the development of a comprehensive SEM characterization methodology, including both preparation and imaging challenges. The samples were designed as multilayers based on Ti and V, consisting of three single-component metallic layers arranged alternately. Elements with similar atomic numbers ($Z_{Ti} = 22$ and $Z_V = 23$) were chosen in order to make a deliberate complication of the analysis, as contrast in SEM microscopy is closely related to atomic number. The top and bottom Ti layers had the same thickness (200 nm), while the thickness of the V middle layer was 100 nm, 50 nm, 30 nm, 20 nm, 10 nm, and 5 nm, respectively. Various thicknesses

of the middle layer were supposed to allow the determination of the resolution limit of the microscope as well as the disadvantages of the preparation technique.

### 2.2. Manufacturing of Multilayer Coatings

Thin-film materials were prepared by pulsed DC magnetron sputtering. We described a detailed description of the applied sputtering method elsewhere [51–59]. Ti/V/Ti multilayers with the desired thickness were obtained by alternately sputtering (with the appropriate power) of targets made of Ti and V. For all prepared multilayers, the deposition processes were carried out in argon plasma at a pressure of $1.2 \times 10^{-2}$ mbar, which was obtained with an argon flow of approximately 26 mL/min. The sputtered materials had the form of metallic titanium and vanadium targets (diameter: 30 mm, thickness: 3 mm, purity: 99.995%). They were alternately sputtered using two individual magnetrons that were powered with adequate power. The supply parameters were selected to take into account differences in the deposition rate of titanium as compared to vanadium. The distance between the target and the substrates ($SiO_2$ and Si) on the rotary drum was 90 mm. The deposition time of the bottom and top Ti films was 20 min., while the deposition time of the V middle layer was related to the desired thickness, which was 30 s up to 7 min. for 5 nm and 100 nm, respectively. Detailed data on the technological parameters of the sputtering processes are collected in Table 1. The thickness of individual films was estimated on SEM images with the use of the tools available in the software. The test samples from the deposition of single Ti and V layers allowed us to estimate the sputtering rate of both materials. Therefore, it was possible to accurately determine the time needed to obtain the desired thickness of the individual layers. These results were verified with the aid of an optical profiler (Talysurf CCI from Taylor Hobson).

**Table 1.** Deposition parameters of Ti/V/Ti multilayers by pulsed DC magnetron sputtering with their thickness.

| | **Deposition Parameters of Ti/V/Ti Multilayers** | | | | | | | | |
|---|---|---|---|---|---|---|---|---|---|
| $P_{Ar}$ [mbar] | **Bottom Ti Layer -Target Ti** | | | **Middle V Layer -Target V** | | | **Top Ti Layer -Target Ti** | | |
| | **Power [W]** | **Time [min.]** | **t [nm]** | **Power [W]** | **Time [min.]** | **t [nm]** | **Power [W]** | **Time [min.]** | **t [nm]** |
| $1.2 \times 10^{-2}$ | 400 | 20 | 200 | 420 | 7 | 100 | 400 | 20 | 200 |
| | | | | | 3.5 | 50 | | | |
| | | | | | 2 | 30 | | | |
| | | | | | 1.5 | 20 | | | |
| | | | | | 1 | 10 | | | |
| | | | | | 0.5 | 5 | | | |

Designations: t—thickness, $P_{Ar}$—pressure of argon during sputtering.

### 2.3. Preparation Techniques and Details of SEM Measurements

For microscopic visualization of multilayers, three different cross section preparation techniques were used: (1) standard layer fracture, (2) FIB with a gallium ion beam, and (3) PFIB (plasma focused ion beam) with xenon plasma. A diagram of the following stages of the preparation techniques used and their analysis by SEM is shown in Figure 1. In addition to microscopic visualization, the analysis of elemental composition was also performed using EDS. Both the cross sections and their imaging were realized using a dualbeam microscope equipped with an electron column for imaging slides and an ion column for micromachining, respectively. All examined samples were glued to the SEM table using copper tape: (i) double-sided from their bottom and (ii) single-sided along the edge of the sample in order to ensure proper charge dissipation and stable mechanical connection.

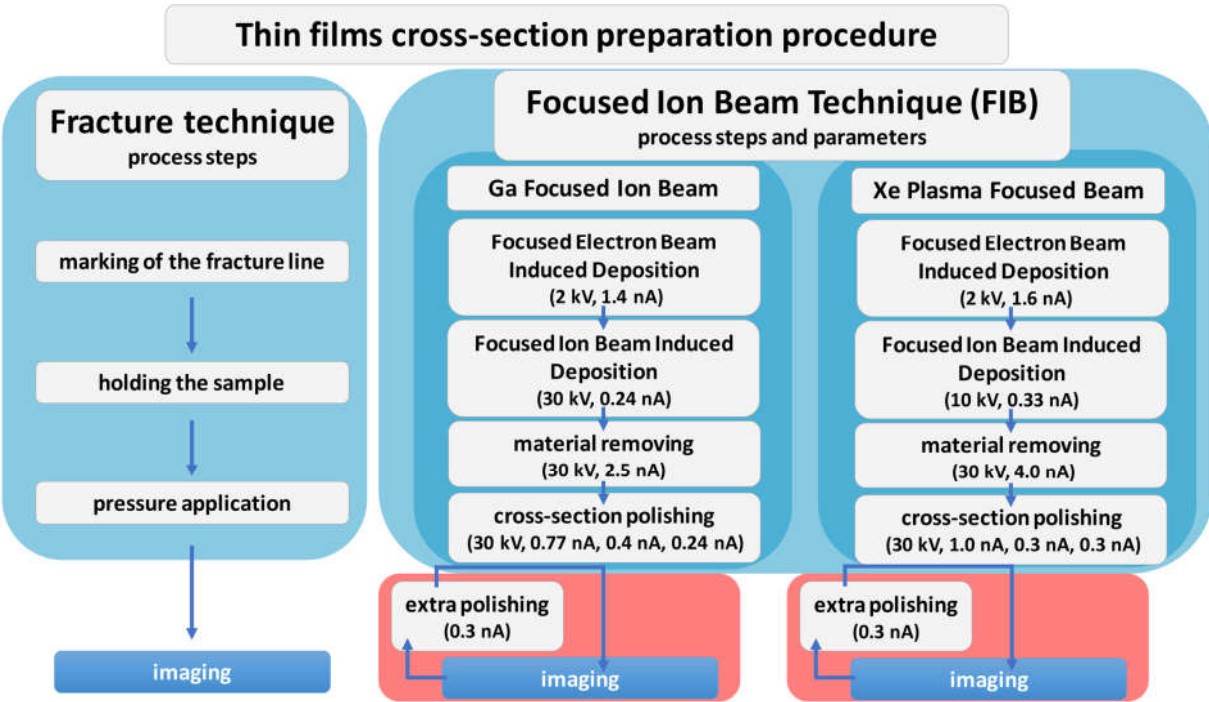

**Figure 1.** Cross-sectional preparation and SEM imaging procedure of thin-film multilayers.

**Fracture of the sample:** in this procedure, the line of the breakthrough was marked on the sample (from a side of the silicon substrate) with a diamond stylus. Then, one end of the specimen was held, while the other was pressed with a laboratory spatula (perpendicular to the surface), causing a break along the plotted line. The breakthroughs prepared in this way were placed in an SEM holder in the form of a vise, allowing imaging perpendicular to the plane of the breakthrough, and ensuring proper charge dissipation and mechanical stability. For their microscopic visualization, a Helios NanoLab 600i SEM microscope was used equipped with a Schottky gun, with a claimed resolution of 1.4 nm @ 1 kV. The imaging was carried out in immersion mode, using a TLD detector, at an acceleration voltage of 2 kV and a current of 0.17 nA.

Preparation of cross sections by focused beam techniques:

- **The preparation of FIB/Ga** was carried out with the aid of a Helios NanoLab 600i microscope. Its ion column is equipped in a gallium ion source Ga/LIMS (liquid metal ion source) with the following parameters: (i) current range 0.1–65 nA, (ii) accelerating voltage: 500 V–30 kV. For their microscopic visualization, Helios NanoLab 600i SEM microscope, equipped with a Schottky gun, with a claimed resolution of 1.4 nm @ 1 kV. The imaging was carried out in immersion mode, using a TLD detector, at an acceleration voltage of 2 kV and a current of 0.17 nA.
- **The preparation of PFIB/Xe** was carried out with the aid of the Helios G4 PFIB CXe microscope. The second source is inductively coupled Xe+ plasma with the following parameters: (i) current range: 1 pA–2.5 uA, (ii) accelerating voltage: 2–30 kV. Its electron column contains a Schottky gun with a claimed resolution of 0.6 nm @ 2–15 kV. Imaging was carried out in immersion mode, using a TLD detector, at an accelerating voltage of 2 kV and a current of 0.1 nA.

The first step in both techniques was selection of the area of interest (AOI), which must be protected from the destructive effects of the ion beam. For this purpose, a protective layer of platinum was applied by focused electron beam-induced deposition (FEBID) and focused ion beam-induced deposition (FIBID). In the FIBID process, ion beam bombardment of the surface results in damage to the surface and thus loss of information from the first layers of an examined sample. The use of the FEBID process, as a primary deposition of the Pt

layer, offsets this problem. Electrons, compared to ions, have negligible mass, so the Pt application was nondestructive. The dimensions of the Pt layers were (i) 10 × 1 × 0.3 μm (FEBID) and (ii) 10 × 1 × 1 μm (FIBID). In the case of FIB/Ga, the beam parameters were as follows: (i) 2 kV and 1.6 nA, (ii) 30 kV and 0.24 nA, respectively. After the deposition of the protective layers, the next step was to remove the pre-AOI material using FIB. The width of the trench is usually equal to the width of the platinum layer, and the depth depends on the expected thickness of the layers, while the length is chosen so that the deepest layers can be observed. In the case of the tested samples, the trench dimensions were 10 × 3 × 1.5 μm, while the parameters of the FIB/Ga beam were 30 kV, 2.5 nA. The final step in preparing the cross section was polishing its face to obtain a smooth section surface. This step requires several iterations, each with a smaller beam current. In this case, it was 0.77 nA, 0.4 nA, and 0.23 nA, with an acceleration voltage of 30 kV. An analogous procedure was used for a xenon plasma microscope. The differences are in the beam parameters. FEBID platinum was applied with beam parameters at 2 kV and 1.6 nA, while FIBID was deposited at 12 kV and 0.33 nA. The grinding was carried out with a beam-accelerating voltage of 30 kV and a current of 4 nA. Double cross-sectional polishing was performed with 30 kV, while the current was set at 1 nA and 0.3 nA, respectively.

It should also be noted that an important additional step in the developed visualization procedure was additional polishing. We have noticed that for samples with low material contrast, it is necessary to polish after the acquisition of each individual SEM image. It is related to the formation of impurities on the surface of the sample as a result of its interaction with the electron beam. Therefore, each passage of the beam on the cross-sectional surface causes a decrease in contrast. For materials with a similar (especially low) atomic number such as titanium and vanadium, where the material contrast is initially low, the additional reduction in the contrast significantly hinders the differentiation of the layers. Even if only a section of the sample was imaged, the contrast will be reduced over the entire cross section. In the case of examined thin-film multilayers, even a small decrease in contrast significantly affects the ability to distinguish layers. Moreover, changes in contrast mean that individual SEM images cannot be truly compared with each other, especially in the context of distinguishing elements based on their atomic number.

## 3. Results

In Figure 2, a comparison of the SEM and EDS measurements of the Ti/V/Ti multilayer cross sections prepared by the fracture technique, FIB/Ga and PFIB/Xe, can be seen. The thickness of the middle V layer was 100 nm, 30 nm, and 10 nm, respectively. As can be seen, depending on the preparation method, the visualization of the sample changes significantly. Only in the case of a fracturing procedure has the microstructure of thin films been preserved, and their columnar character can be seen. However, it is difficult to determine the position in which the base and top of the columns are located. Therefore, it is often impossible to determine even the position of the middle vanadium layer. Similar results can be seen in such works as [15,19,28]. In our studies, the columns were sometimes randomly broken off at different heights, regardless of the layered structure of the coating. In the case of the thinnest sample, the individual columns end up where the vanadium layer was located. This effect is similar to the renucleation of a layer after an interrupted deposition process. The column widths for all Ti/V/Ti multilayers are comparable. It is possible to precisely determine their widths of 39 ÷ 50 nm for the sample with 100 nm of the middle V layer, as well as 44 ÷ 56 nm for the 30 nm of the V layer, and 23 ÷ 62 nm for the coating with 10 nm of V layer (Figure 3). The most important limitation of the fracture method is that the cross section does not show the material contrast, and it is impossible to distinguish all individual layers.

## Thin-films cross-section preparation technique

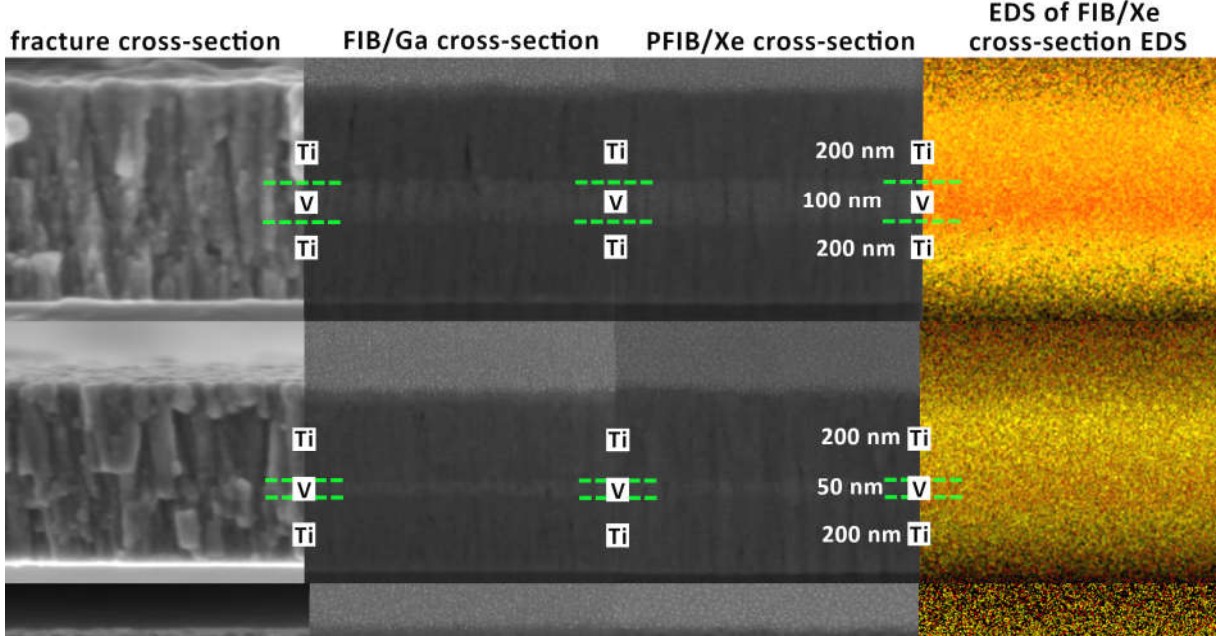

**Figure 2.** Cross sections of Ti/V/Ti multilayers with various thicknesses of middle V layer (100, 30, and 10 nm) based on SEM and EDS measurements, prepared by fracture technique, FIB/Ga, and PFIB/Xe. Note: The SEM images were recorded at high resolution and their original version can be found in the Supplementary Materials.

The results obtained with a focused ion beam have a different appearance. Both techniques, PFIB/Xe and FIB/Ga, provide high material contrast, and it is possible to distinguish Ti and V layers. However, in both cases, the thickness of the central V layer of 30 nm can be assumed as the limit. For the 10 nm V layer, a slight change in material contrast is apparent, but it is difficult to delineate it precisely. This layer could easily be overlooked, especially when the structure of the multilayer under examination is unknown. It is difficult to relate these results to the literature, as no work has been encountered on FIB cross sections through thin films with similar atomic numbers. In the case of FIB preparation, the main problem is that information about the microstructure of the analyzed coating is mostly lost. Only the outlines of individual columns and the spaces between columns can be identified. Comparison of the structure of the PFIB/Xe and FIB/Ga cross sections indicates a better distinction of the columns based on the PFIB/Xe method. This may be related to the formation of a thinner amorphous layer due to the use of Xe ions compared to Ga ions [1,60–62]. Determining the width of the columns is also possible in the case of both FIB preparation techniques, but it is significantly more difficult than the fracturing technique. For PFIB/Xe, the measured widths were 44 ÷ 46 nm, 35 ÷ 45 nm, and 46 ÷ 45 nm for multilayers with a middle V layer of 100 nm, 30 nm, and 10 nm, respectively (Figure 3). However, for FIB/Ga, these widths were 41 ÷ 49 nm, 46 ÷ 52 nm, and 34 ÷ 44 nm, respectively (Figure 3). These results are in good agreement with those obtained by the fracture method. However, a wider range of measured column widths can be observed in samples prepared by FIB/Ga. Thus, it can be concluded that both cross sections made with a focused ion beam have similar characteristics. The apparent

differences related to the preservation of the original microstructure of the sample are relatively insignificant. However, it can be assumed that the best preparation method, burdened with fewer artifacts, is PFIB/Xe.

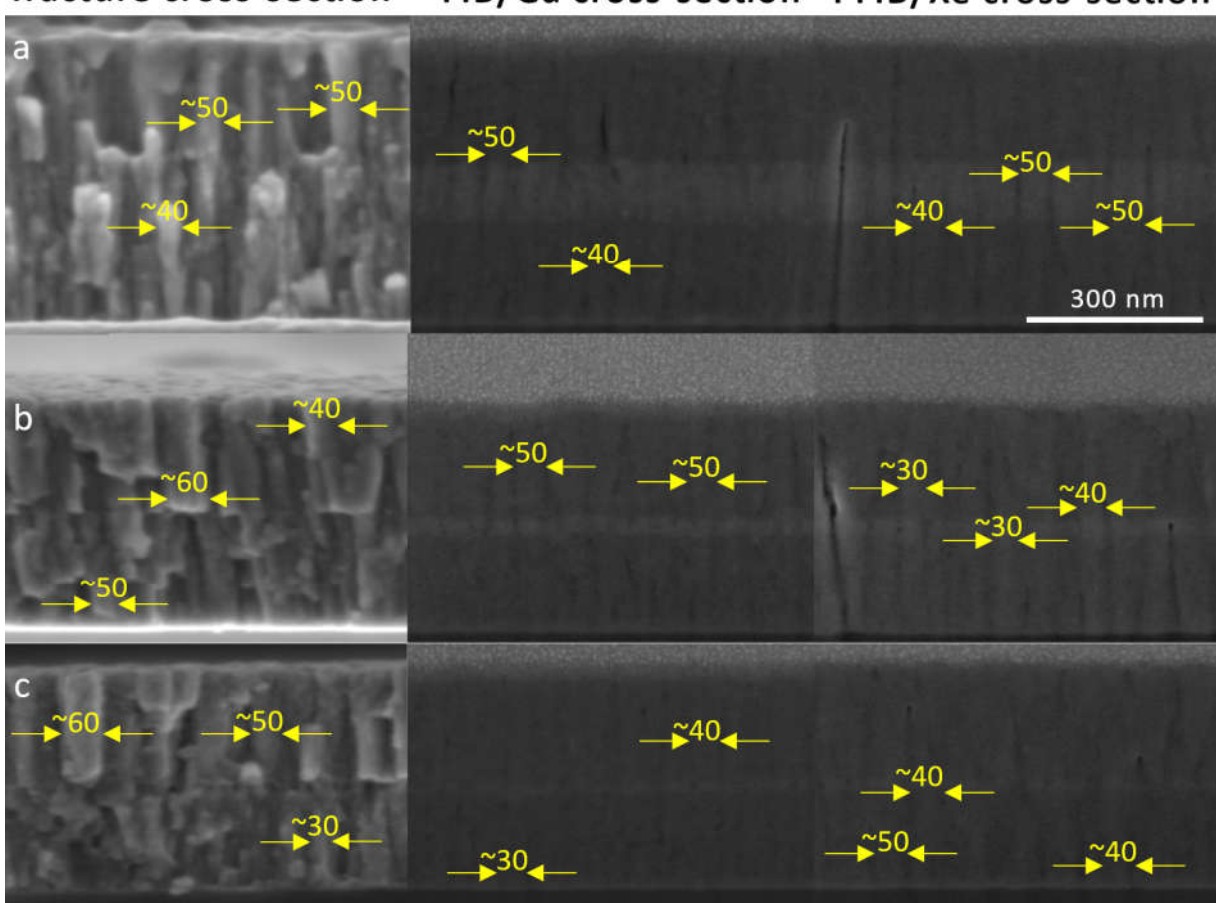

**Figure 3.** Determination of column width based on cross sections of Ti/V/Ti multilayers (with various thicknesses of the middle V layer: (**a**) 100 nm, (**b**) 30 nm, and (**c**) 10 nm) prepared by fracture technique, FIB/Ga, and PFIB/Xe. Note: The SEM images were recorded at high resolution and their original version can be found in the Supplementary Materials.

It should also be mentioned that the undeniable advantage of the FIB/Ga and PFIB/Xe preparation is the possibility of obtaining a high material contrast, and thus the ability to distinguish the material composition of multilayers with slightly different atomic numbers, i.e., $Z_{Ti} = 22$ and $Z_V = 23$. As can be seen in Figure 2, the EDS method does not provide such capabilities. It is impossible to clearly determine the interface between individual layers; thus, the method does not allow one to accurately determine their thickness. Even in the case of a Ti/V/Ti multilayer with the 100 nm middle V layer, where the location of all layers corresponds to their actual arrangement, the exact determination of the vanadium thickness is difficult. The problem worsens as the thickness of the middle layer decreases. For a V thickness of 30 nm, it is possible to speculate with an assumed approximation about the position of the middle layer, but neither its position nor its width corresponds to the SEM images of the cross section. For a V thickness of 10 nm, the signal from both elements is evenly distributed over the cross-sectional area, so it is not possible to observe individual layers. It should be noted that for layers of 10 nm or less, despite the preservation of material contrast, SEM images alone may not be sufficient to define the multilayer structure of the sample. Most of the available literature only reports percentage results of EDS analysis, e.g., [63,64], or elemental maps of the surface, e.g., [65], so it is not possible to

confront obtained results with them. In such a case, EDS analysis is used to determine the elements comprising the sample and complements the FIB section imaging.

The next step of the study was to determine the resolution afforded by the preparation of FIB/Ga and PFIB/Xe methods as compared to the fracturing technique. Therefore, cross sections of multilayers with V layer thicknesses of 100, 50, 30, and 5 nm are included (Figure 4). As can be seen, it is difficult to discern where the V layer is on the cross section resulting from conventional fracturing. The growth of both the titanium and vanadium layers during deposition was similar, so their columnar morphology can be distinguished. In some cases, it is even possible to specify only by random breakage of the layers where columns begin with the next layer, but it is not repetitive and is often ambiguous. This is due to the fact that with a small thickness of the films included in the multilayer coating, the effect of reproducing the growth of the previous lower layer occurs (Figure 4a). The situation is completely different in the case of a breakthrough obtained with both FIB techniques, where the 10 nm thick multilayer is the last distinguishable one (Figure 4b,c). Identification of a 5 nm thick layer with an unknown sample structure is impossible. It can be possible only for well-defined and specially designed multilayers, most often as a comparison of SEM images of several samples. The difference in contrast between such thin films is imperceptible. As can be seen in Figure 4, the resolution limit has been reached. Moreover, it can be noticed that the material contrast for cross sections obtained by the PFIB/Xe and FIB/Ga methods is similar. The disclosed microstructures of these two cross sections are convergent. However, the columnar nature of the multilayer was better revealed in the case of PFIB/Xe. Most probably, it is related to the implantation of gallium ions in the structure of the Ti/V/Ti cross sections [1,40,41]. This effect is known [42] and while its influence on tested materials can sometimes be neglected, in the case of the multilayers analyzed, the result of fewer artifacts results from the PFIB/Xe preparation technique.

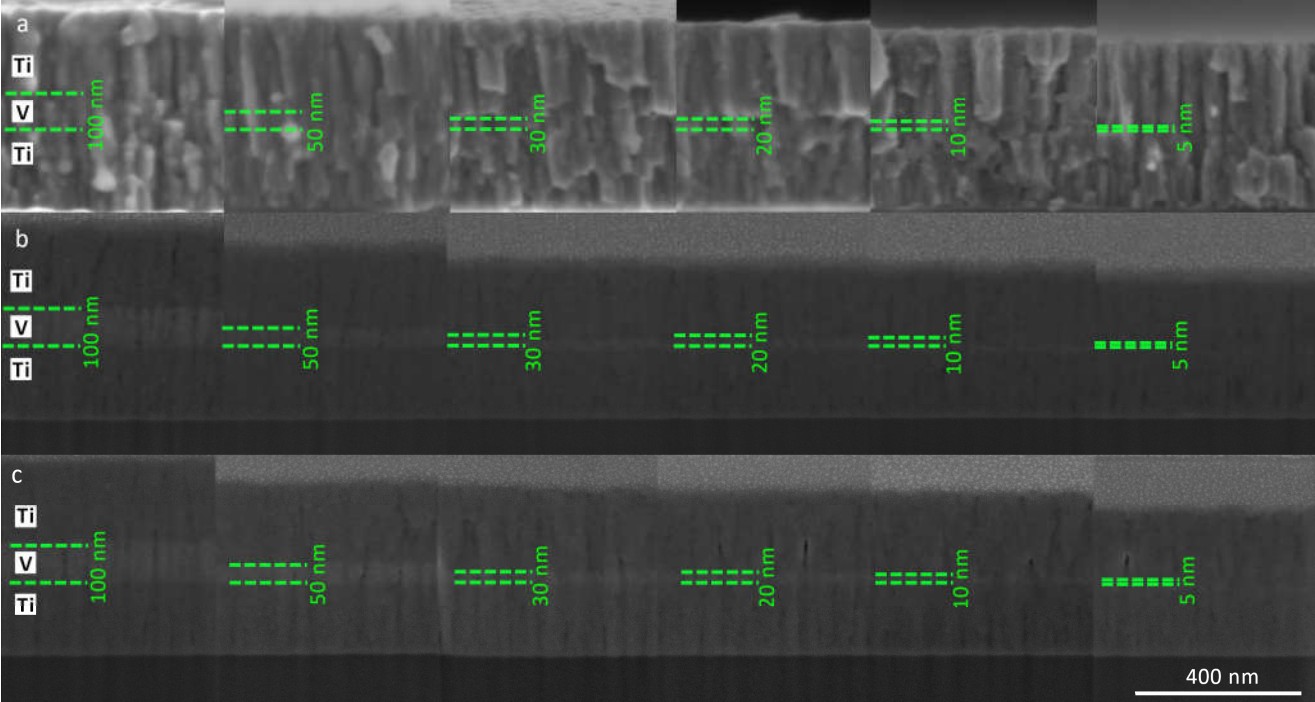

**Figure 4.** Comparison of the resolution afforded by the preparation of Ti/V/Ti multilayer cross sections by: (**a**) fracturing, (**b**) FIB/Ga, (**c**) PFIB/Xe, where the thickness of the middle V layer was reduced from 100 nm to 5 nm. Note: SEM images were recorded in high resolution, and their original version can be found in Supplementary Materials.

## 4. Conclusions

An improved methodology for the SEM study of thin-film coatings as an innovative multithreaded approach to the preparation itself was described. It allows information to be retained about the actual microstructure and ensures high material contrast even for elements with similar atomic numbers. It was found that determining the microstructure application of the standard fracture technique is necessary (as with most of the noninvasive methods). However, while the microstructure was preserved, the material contrast remained invisible, making it impossible to distinguish each layer in the construction of a multilayer. EDS analysis was also not sufficient to present their construction and the interfaces were not clearly defined. Even when the thickness of individual V films was around 100 nm, it was difficult to determine by EDS the position of the interfaces between individual layers in a Ti/V/Ti multilayer. Material contrast enhancement occurs only for FIB techniques. Cross-sectional studies showed that the 10 nm mid-V layer was the resolution limit. Moreover, a comparison of the PFIB/Xe and FIB/Ga cross sections revealed that fewer artifacts give the PFIB method, and hence this technique seems to be better for the analysis of multilayer nanostructures. In our opinion, reliable information about the properties of modern nanomaterials, especially multilayers used in electronics, can be obtained by analyzing a two-part SEM image, where the first is a fracture, while the second is a FIB/PFIB cross section. It is worth noting that there were only subtle differences between SEM images of cross sections made by FIB-Ga and PFIB-Xe, but the decrease in local amorphization, the lack of Ga-ions incorporation into the sample, and slightly better contrast are in favor of Xe plasma.

**Supplementary Materials:** The following supporting information can be downloaded at: https://www.mdpi.com/article/10.3390/coatings13020316/s1, Figure S1: Cross-sectional preparation and SEM imaging procedure of thin-film multilayers.; Figure S2: Cross sections of Ti/V/Ti multilayers with various thicknesses of middle V layer (100, 30, and 10 nm) based on SEM and EDS measurements, prepared by fracture technique, FIB/Ga, and PFIB/Xe. Figure S3: Determination of column width based on cross sections of Ti/V/Ti multilayers (with various thicknesses of the middle V layer: (**a**) 100 nm, (**b**) 30 nm, and (**c**) 10 nm) prepared by fracture technique, FIB/Ga, and PFIB/Xe. Figure S4: Comparison of the resolution afforded by the preparation of Ti/V/Ti multilayer cross sections by: (**a**) fracturing, (**b**) FIB/Ga, (**c**) PFIB/Xe, where the thickness of the middle V layer was reduced from 100 nm to 5 nm.

**Author Contributions:** Conceptualization, D.W. and M.S.; methodology, M.S. and D.W.; validation, D.W.; investigation, M.S. and A.C.; resources, A.Z.; writing—original draft preparation, M.S. and D.W.; writing—review and editing, M.S. and D.W.; supervision, D.W. All authors have read and agreed to the published version of the manuscript.

**Funding:** This work was financed from the sources given by the MNiSW in the years 2020–2024 as Implementation Doctorate Programme (project number: DWD/4/5/2020).

**Institutional Review Board Statement:** Not applicable.

**Informed Consent Statement:** Not applicable.

**Data Availability Statement:** The data presented in this study are available on request from the corresponding author.

**Conflicts of Interest:** The authors declare no conflict of interest.

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
