# Peer review of "Improved Methodology of Cross-Sectional SEM Analysis of Thin-Film Multilayers Prepared by Magnetron Sputtering"

_coatings, doi:10.3390/coatings13020316_

Round 1
Reviewer 1 Report
Title: Improved Methodology of Cross-sectional SEM Analysis of Thin-film Multilayers prepared by Magnetron Sputtering
Authors: Malwina Sikora, Damian Wojcieszak, Aleksandra Chudzynska, and Aneta Zeiba
The manuscript entitled “Improved Methodology of Cross-sectional SEM Analysis of Thin-film Multilayers prepared by Magnetron Sputtering” by Malwina Sikora et al. presents the various methodology for sample preparation of cross-sectional scanning electron microscopy imaging and analysis. The samples for cross-section SEM images are prepared by three different methodologies 1) standard fracture, 2) focused ion beam, and 3) plasma-focused ion beam, and the results are compared. The manuscript could be recommended for publication with major modifications.
Q1: Authors present three different methodologies for the sample preparation cross-sectional SEM analysis. If so, why did they choose the multilayer coating? What is the advantage of selecting a Ti/V/Ti multilayer system with very low electron density contrast at the interface? It should be emphasized in the introduction section.
Q2: Sample preparation using a focused ion beam for cross-sectional SEM and TEM is a well-known method. Whether the authors improve the methodology to image an interface with a shallow electron density contrast? What is new in this manuscript?
Q3: Why do the authors represent low-quality SEM images for all four figures? SEM is a powerful imaging technique, and authors should check the literature and compare it.
Q4: The main finding of this manuscript is the high-resolution SEM images. There are no high-resolution SEM images in the manuscript. Most images are fully back, and layer structures are not seen in all the figures. How do authors justify the results with an improved methodology? Is the method improved?
Author Response
Answers to the report of Reviewer
on the manuscript entitled: Improved Methodology of Cross-sectional SEM Analysis of Thin-film Multilayers prepared by Magnetron Sputtering
Authors: Malwina Sikora, Damian Wojcieszak, Aleksandra Chudzynska, and Aneta Zieba
Authors :
We express our gratitude for your remarks that allowed us improve our manuscript. We have taken them into account in the revised version of my paper. Answering to the reviewer’s remarks, we have introduced some revisions to the manuscript. English language correction was made in the text.
- Reviewer 1:
Authors present three different methodologies for the sample preparation cross-sectional SEM analysis. If so, why did they choose the multilayer coating? What is the advantage of selecting a Ti/V/Ti multilayer system with very low electron density contrast at the interface? It should be emphasized in the introduction section.
- Authors:
The multilayer coating was chosen intentionally, due to the difficulty in imaging and identifying single layers in their structure. The use of elements with similar atomic number and low contrast was intended to present a challenge in terms of SEM and EDS analysis. In the practice of SEM work, there are non-obvious samples whose identification requires the multi-step approach presented in this paper. FIB techniques offer excellent possibilities, but they are not without flaws, for example related to the change in the structure of the excised specimen (including its delicate amorphization). By undertaking research on samples with strictly known composition and structure, it was possible to develop a method for the analysis of complex multilayer coatings.
Selection of the appropriate preparation technique, even from among the already recognized methods, requires knowledge of how each of them changes the actual properties of the sample and causes artifacts. Making such a seemingly simple choice is not easy due to the very small number of studies that compare the results of different methods of preparation of the same sample. In the case of modern nanomaterials, this is additionally difficult because the parameters describing them are often statistical in nature, such as the average size of crystallites. For this reason, a good research material for a reliable comparison of several methods is multilayer coating, where well-defined, single layer of the desired thickness can be buried at a specific and known depth. It is worth noting that a series of multilayers with a gradually modified thickness of selected layers allows for better visibility of quite subtle differences and artifacts that could have arisen, especially in the interface area. Another issue that makes difficult to choose the optimal preparation is related to the fact that most of publications refer to the advantages and disadvantages of only one method, and unfortunately often without any explanation of the reasons for its choice. The third issue to be mentioned is the use of quite contrasting materials (for SEM), which will have a rather poor application for materials consisting of elements with a similar atomic number. Their use results in a very low electron density contrast at the interface and is an additional difficulty in SEM imaging, like the titanium and vanadium selected for our research, but will allow for the elimination of much more subtle artifacts.
According to the comment, Section Introduction was improved and extended as follows:
“Selection of the appropriate preparation technique, even from among the already recognized methods, requires knowledge of how each of them changes the actual properties of the sample. Making such a seemingly simple choice is not easy due to the very small number of studies that compare the results of different methods of preparation of the same samples [15-16]. In the case of modern nanomaterials, this is also difficult because the parameters that describe them are often statistical in nature, such as the average size of the crystallites [17]. For this reason, a good research material for a reliable comparison of several methods is multilayer coating, where a well-defined single layer of the desired thickness can be buried at a specific and known depth [18-23]. It should be noted that a series of multilayers with a gradually modified thickness of selected layers allows for better visibility of quite subtle differences and artifacts that could have arisen, especially in the interface area [24-26]. Another issue that makes it difficult to choose the optimal preparation is related to the fact that most publications refer to the advantages and disadvantages of one single method, and unfortunately often without any explanation of the reasons for its choice [27-33]. The third issue to be mentioned is the use of quite contrasting materials (for SEM), which will have a rather poor application for materials consisting of elements with a similar atomic number [34-35]. Their use results in a very low electron density contrast at the interface and is an additional difficulty in SEM imaging (like the titanium and vanadium selected for our research), but will allow for the elimination of much more subtle artifacts.”
- Reviewer 1:
Sample preparation using a focused ion beam for cross-sectional SEM and TEM is a well-known method. Whether the authors improve the methodology to image an interface with a shallow electron density contrast? What is new in this manuscript?
- Authors:
We agree with the reviewer's comment that the FIB method is known and widely used. However, our goal was to develop an improved methodology for study thin-film coatings based on known and existing methods, including the aforementioned FIB. Our innovative approach does not concern the mere improvement of any of the methods, but it is devoted to a multi-threaded approach to the SEM preparation itself, which will allow to retain information about the actual microstructure and ensure the best possible material contrast (even for elements with similar atomic numbers). The research we performed on the example of Ti/V/Ti multilayers, in order to present the sense of using our improved methodology, showed that the true information about the tested sample can be obtained by assessing its microstructure based on SEM images made using the fracture technique, while the material composition can be well visualized using FIB methods (with better resolution than EDS). In our opinion, reliable information about the properties of modern nanomaterials, especially multilayers used in electronics, can be obtained by analyzing a two-part SEM image, where the first one is a fracture, while the second is a FIB cross-section. It is worth noting that there are only subtle differences between SEM images of cross sections made by FIB-Ga and PFIB-Xe, but decrease of local amorphization, lack of Ga-ions incorporation into the sample, or slightly better contrast are in favor of Xe plasma. Literature review also indicate on the lack of data describing the application of such methodology to the analysis of a multilayer coatings and, in particular, its effect on their structure. Comprehensive studies comparing the use of different techniques for cross section preparation, especially using two different sources of focused ion beam (including Focused Xenon Plasma) are also being omitted. So, the presented manuscript fills this niche.
In response to the reviewer's remark, we stated that the aim of our work and the intention to use our improved methodology should be emphasized in the article. According to the comment, the manuscript was corrected/extended as follows:
Section Abstract:
“In this work an improved methodology of cross-sectional Scanning Electron Microscopy (SEM) analysis of thin-film Ti/V/Ti multilayers was described. Multilayers with various thicknesses of the vanadium middle layer were prepared by magnetron sputtering. The differences in cross sections made by standard fracture, Focused Ion Beam (FIB)/Ga, and Plasma Focused Ion Beam (PFIB)/Xe have been compared. For microscopic characterization the Helios NanoLab 600i microscope and the Helios G4 CXe with the Quanta XFlash 630 energy dispersive spectroscopy detector from Bruker were used. The innovative multi-threaded approach to the SEM preparation itself, which allows us to retain information about the actual microstructure and ensure high material contrast even for elements with similar atomic numbers was proposed. The fracture technique was the most noninvasive for microstructure, whereas FIB/PFIB results in better material contrast (even than EDS). There were only subtle differences in cross sections made by FIB-Ga and PFIB-Xe, but the decrease in local amorphization or slightly better contrast was in favor of Xe plasma. It was found that reliable information about the properties of modern nanomaterials, especially multilayers, can be obtained by analyzing a two-part SEM image, where the first one is a fracture, while the second is a PFIB cross section.”
Section Introduction:
“Therefore, it seems sufficient to visualize various types of advanced electronic, photonic, or optical systems based on nanostructures. However, there are still many problems with proper preparation of the samples for SEM. It should be noted that while there is no ideal method that would be suitable for every sample, an improved methodology based on the hybrid preparation technique as a multistep approach can be applied as a modern solution.
Improvement of the SEM result requires the use of such cross section preparation technique that will maintain the real properties of individual layers and interfaces [16; 37]. There are a number of methods for manufacturing cross sections of thin films. Among these are break [37-39], pre-cut technique [40], ultramicrotomy [37, 41, 42], grinding and polishing preceded by resin encapsulation [37], as well as ionic techniques: ion polishing [43, 44, 45] and focused ion beam [46-47]. In each method, there are artefacts that affect the visualization of the properties of the samples, which one needs to be able to dissect. In the literature on thin films, the field of preparation is usually ignored; only SEM images of cross sections are presented. Many works such as [eg. 48, 49] show results obtained by only one technique, which usually is standard fracture. There is a lack of summaries comparing different preparation methods with each other, and this paper is an attempt to address this niche. In our opinion, the improved methodology for the manufacturing of thin-film preparations for the purposes of SEM research can be successfully implemented using three methods, which are fracture and Focused Ion Beams with Gallium ion source and Xenon plasma.
…….
It should be noted that in many works (e.g., [36]) dedicated to SEM studies of thin film materials, especially prepared by PVD or CVD methods) the methodology for preparation techniques for accurate microscopic visualization is not given or has a residual description. This causes difficulties in the proper interpretation and characterization. Hence, arises the need to develop a complex methodology. The aim of our work was to develop an improved methodology for the study of thin-film coatings on the basis of known and existing methods, including the aforementioned FIB. This innovative approach does not concern the improvement of standard methods, but it is devoted to a multithreaded approach to the SEM preparation itself, which will allow us to retain information about the actual microstructure and ensure the best possible material contrast (even for elements with similar atomic numbers). The research performed on the example of Ti/V/Ti multilayers, in order to present the sense of using such an improved methodology, showed that the true information about the tested sample can be obtained by assessing its microstructure based on SEM images made using the fracture technique, while the material composition can be well visualized using FIB methods (better than by EDS). The literature review also indicates the lack of data that describe the application of such methodology to the analysis of multilayer coatings and, in particular, its effect on their structure. Comprehensive studies comparing the use of different techniques for cross-section preparation, especially using two different sources of focused ion beam (including Focused Xenon Plasma) are also being omitted. Therefore, this work fills that niche.”
Section Conclusions
“An improved methodology for the SEM study of thin-film coatings as an innovative multithreaded approach to the preparation itself was described. It allows information to be retained about the actual microstructure and ensures high material contrast even for elements with similar atomic numbers. It was found that to determine the microstructure application of the standard fracture technique is necessary (as most of the noninvasive). However, while the microstructure was preserved, the material contrast remained invisible, making it impossible to distinguish each layer in the construction of a multilayer. EDS analysis was also not sufficient to present their construction and the interfaces were not clearly defined. Even when the thickness of individual V films was around 100 nm, it was difficult to determine by EDS the position of the interfaces between individual layers in a Ti/V/Ti multilayer. Material contrast enhancement occurs only for FIB techniques. Cross-sectional studies showed that the 10 nm mid-V layer was the resolution limit. Moreover, a comparison of the PFIB/Xe and FIB/Ga cross sections revealed that fewer artifacts give the PFIB method, and hence this technique seems to be better for the analysis of multilayer nanostructures. In our opinion, reliable information about the properties of modern nanomaterials, especially multilayers used in electronics, can be obtained by analyzing a two-part SEM image, where the first is a fracture, while the second is a FIB/PFIB cross section. It is worth noting that there were only subtle differences between SEM images of cross sections made by FIB-Ga and PFIB-Xe, but the decrease in local amorphization, the lack of Ga-ions incorporation into the sample, or slightly better contrast are in favor of Xe plasma.”
- Reviewer 1:
Why do the authors represent low-quality SEM images for all four figures? SEM is a powerful imaging technique, and authors should check the literature and compare it.
The main finding of this manuscript is the high-resolution SEM images. There are no high-resolution SEM images in the manuscript. Most images are fully back, and layer structures are not seen in all the figures. How do authors justify the results with an improved methodology? Is the method improved?
- Authors:
The authors agree with the reviewer's comment that the quality of SEM images should be high and it is not difficult to ensure it. However, the problem arises from the fact that the journal's editing system drastically reduces the resolution of the images when creating a pdf file with the manuscript (sent for review). For this reason, the authors have accommodated the high-resolution images separately in the system. To eliminate this problem in an additional way, these high-resolution images will be added as supplementary materials. Additional information was added to figure captions as follows:
“Note: SEM images were recorded in high resolution and their original version can be found in supplementary materials.”
We would like to add that when the quality of the images of the investigated structures is not clear to determine, the visibility of platinum grains on the cross-section is used as a determinant. In all of the cross-sectional images provided, platinum grains are visible, which indicates their high quality. As for the brightness of the SEM images, due to the small difference with the atomic number, it cannot be significantly higher (especially when the middle layer is 10 nm thick). However, these rather subtle differences between FIB and PFIB will be more visible in the original quality images (included as mentioned supplementary materials).

Reviewer 2 Report
In this manuscript, to determine the influence of fabrication techniques on the visualization of the properties of cross-sections multilayer coatings cross-sections by SEM, the differences in cross-sections made by standard fracture, Focused Ion Beam (FIB)/Ga and Plasma Focused Ion Beam (PFIB)/Xe were compared. This work should be of interest to the readership of Coatings. The authors may wish to consider the following comments in a revised version:
1. The introduction part should be well-organized. The 3rd paragraph was too long to show the novelty of this work.
2. Line 238, “The In the Figure 2 comparison of SEM and EDS measurements of Ti/V/Ti multilayers cross-sections prepared by fracture technique, FIB/Ga and PFIB/Xe can be seen” This sentence should be revised.
3. The SEM images presented in this study were fuzzy. Images with high quality are suggested.
Author Response
Answers to the report of Reviewer
on the manuscript entitled: Improved Methodology of Cross-sectional SEM Analysis of Thin-film Multilayers prepared by Magnetron Sputtering
Authors: Malwina Sikora, Damian Wojcieszak, Aleksandra Chudzynska, and Aneta Zieba
Authors :
We express our gratitude for your remarks that allowed us improve our manuscript. We have taken them into account in the revised version of my paper. Answering to the reviewer’s remarks, we have introduced some revisions to the manuscript. English language correction was made in the text.
- Reviewer 2:
The introduction part should be well-organized. The 3rd paragraph was too long to show the novelty of this work.
- Authors:
According to the comment section introduction was rewritten. the most important changes are marked in red. Mentioned paragraph has been significantly shortened and part of it separated as a separate next paragraph.
The Section introduction was changed as follows:
,,The smaller the area of interest, the more demanding sample preparation is, not to mention the need to reduce artifacts and unwanted modification of the sample itself. Selection of the appropriate preparation technique, even from among the already recognized methods, requires knowledge of how each of them changes the actual properties of the sample. Making such a seemingly simple choice is not easy due to the very small number of studies that compare the results of different methods of preparation of the same samples [15-16]. In the case of modern nanomaterials, this is also difficult because the parameters that describe them are often statistical in nature, such as the average size of the crystallites [17]. For this reason, a good research material for a reliable comparison of several methods is multilayer coating, where a well-defined single layer of the desired thickness can be buried at a specific and known depth [18-23]. It should be noted that a series of multilayers with a gradually modified thickness of selected layers allows for better visibility of quite subtle differences and artifacts that could have arisen, especially in the interface area [24-26]. Another issue that makes it difficult to choose the optimal preparation is related to the fact that most publications refer to the advantages and disadvantages of one single method, and unfortunately often without any explanation of the reasons for its choice [27-33]. The third issue to be mentioned is the use of quite contrasting materials (for SEM), which will have a rather poor application for materials consisting of elements with a similar atomic number [34-35]. Their use results in a very low electron density contrast at the interface and is an additional difficulty in SEM imaging (like the titanium and vanadium selected for our research), but will allow for the elimination of much more subtle artifacts.
….
However, there are still many problems with proper preparation of the samples for SEM. It should be noted that while there is no ideal method that would be suitable for every sample, an improved methodology based on the hybrid preparation technique as a multistep approach can be applied as a modern solution.
………
There are a number of methods for manufacturing cross sections of thin films. Among these are break [37-39], pre-cut technique [40], ultramicrotomy [37, 41, 42], grinding and polishing preceded by resin encapsulation [37], as well as ionic techniques: ion polishing [43, 44, 45] and focused ion beam [46-47]. In each method, there are artefacts that affect the visualization of the properties of the samples, which one needs to be able to dissect. In the literature on thin films, the field of preparation is usually ignored; only SEM images of cross sections are presented. Many works such as [eg. 48, 49] show results obtained by only one technique, which usually is standard fracture. There is a lack of summaries comparing different preparation methods with each other, and this paper is an attempt to address this niche. In our opinion, the improved methodology for the manufacturing of thin-film preparations for the purposes of SEM research can be successfully implemented using three methods, which are fracture and Focused Ion Beams with Gallium ion source and Xenon plasma.
…….
The aim of our work was to develop an improved methodology for the study of thin-film coatings on the basis of known and existing methods, including the aforementioned FIB. This innovative approach does not concern the improvement of standard methods, but it is devoted to a multithreaded approach to the SEM preparation itself, which will allow us to retain information about the actual microstructure and ensure the best possible material contrast (even for elements with similar atomic numbers). The research performed on the example of Ti/V/Ti multilayers, in order to present the sense of using such an improved methodology, showed that the true information about the tested sample can be obtained by assessing its microstructure based on SEM images made using the fracture technique, while the material composition can be well visualized using FIB methods (better than by EDS). The literature review also indicates the lack of data that describe the application of such methodology to the analysis of multilayer coatings and, in particular, its effect on their structure. Comprehensive studies comparing the use of different techniques for cross-section preparation, especially using two different sources of focused ion beam (including Focused Xenon Plasma) are also being omitted. Therefore, this work fills that niche.”
- Reviewer 2:
Line 238, “The In the Figure 2 comparison of SEM and EDS measurements of Ti/V/Ti multilayers cross-sections prepared by fracture technique, FIB/Ga and PFIB/Xe can be seen” This sentence should be revised.
- Authors:
According to the comment, the sentence has been improved.
- Reviewer 2:
The SEM images presented in this study were fuzzy. Images with high quality are suggested
- Authors:
The authors agree with the reviewer's comment that the quality of SEM images should be high and it is not difficult to ensure it. However, the problem arises from the fact that the journal's editing system drastically reduces the resolution of the images when creating a pdf file with the manuscript (sent for review). For this reason, the authors have accommodated the high-resolution images separately in the system. To eliminate this problem in an additional way, these high-resolution images will be added as supplementary materials. Additional information was added to figure captions as follows:
“Note: SEM images were recorded in high resolution and their original version can be found in supplementary materials.”
We would like to add that when the quality of the images of the investigated structures is not clear to determine, the visibility of platinum grains on the cross-section is used as a determinant. In all of the cross-sectional images provided, platinum grains are visible, which indicates their high quality

Round 2
Reviewer 1 Report
The manuscript entitled “Improved Methodology of Cross-sectional SEM Analysis of Thin-film Multilayers prepared by Magnetron Sputtering” by Malwina Sikora et al. presents the various methodology for sample preparation of cross-sectional scanning electron microscopy imaging and analysis. The samples for cross-section SEM images are prepared by three different methods 1) standard fracture, 2) focused ion beam, and 3) plasma-focused ion beam, and the results are compared.
After addressing all the suggestions, the revised manuscript improved and looks good. I will recommend it for publication with the present version.